# Experimenting with Multimodal AutoML: Detection and Evaluation of Alzheimer's Disease

**Ujjawal Shah & Saurav K. Aryal**
Department of Computer Science
Howard University
Washington, DC 20059, USA
`ujjawal.shah@bison.howard.edu & saurav.aryal@howard.edu`

## Abstract

This paper describes an experiment using AutoML, AutoGluon Tabular, to discover multimodal models for MMSE regression and AD detection. Using the ADReSSo dataset, this paper reports enhanced performance in classification models and comparable performance in regression models to the baseline, achieving a significant improvement of up to 82% accuracy on the test dataset. In contrast, the test RMSE has a marginal difference of only 0.28 compared to the baseline.

## 1 Introduction

As modern medicine and preventive healthcare become more efficient and accessible, life expectancy has increased worldwide. As a result, there is a growing need to conduct more research on aging-related illnesses, such as Alzheimer's Disease (AD). Estimates suggest that by 2050 Li et al. (2022); Nichols et al. (2019), around 152 million individuals will be affected by AD and other forms of dementia, leading to increased social and economic costs on families. Therefore, it is crucial to develop technologies to reduce the costs of detecting, evaluating, and tracking AD. This paper will focus on the prediction and scoring of mental status with the AutoML framework AutoGluon-Tabular (Erickson et al., 2020) while utilizing the combination of raw translated text from the *PAR* values, acoustic features and dimension reduction techniques. This paper refers to Luz et al. (2021) as the baseline and compares the performance of our models against it. The baseline uses eGeMAPS feature set Eyben et al. (2016) as acoustic features whereas lexical & morphological descriptions from automatic-transcribed audio files (generated by Google Cloud-based Speech Recogniser) are used as linguistic features. Unlike baseline, which uses acoustic and linguistic features independently for the modeling, this paper instead utilizes acoustic features & raw translated text combinedly for the modeling, also called Multimodal approach.

## 2 Methodology

This paper utilizes the ADReSSo dataset as a basis for comparison, which includes 166 samples in the Train Dataset and 71 samples in the Test Dataset. The dataset is balanced in terms of gender and class and includes normalized audio data. For audio feature extraction, the well-known librosa library McFee et al. (2015) is used, which extracts seven different characteristics, including Mel-Frequency Cepstral Coefficients (MFCCs), Root-mean-square (RMS), Zero Crossing Rate, Tempogram, Spectogram, Spectral Roll, and Spectral Centroid features. The resulting feature arrays are split into individual columns, resulting in 1733 features. Principal Component Analysis (PCA), a linear dimensionality reduction technique, is applied to the acoustic features to avoid the curse of dimensionality. Based on the cumulative variance explained by the principal components as shown in Appendix 2, 50 principal components are used, which explains around 90% of the variance. The intuition behind doing so is to select the best number of principal components while keeping as much of the variance in the original data as possible. Furthermore, the audio data is translated using Google's Text To Speech API and segmented audio files are used for transcriptions due to Google's audio length restrictions. The segmented audio files are split into *PAR*, *INV*, and Both. After

the features are carefully extracted and dimension reduction technique is applied, the acoustic features alongside raw text is fitted to experiment with MultiModal AutoML models for both MMSE regression & AD detection task.

## 3 RESULTS

Compared to the baseline, the paper presents an improvement in the performance of classification models and a similar performance on the regression model. The accuracy score on the test dataset reached to a high of 82% which is a significant jump when compared to the baseline. The test RMSE however has a mere difference of 0.28 with the baseline. The Classification report for the test dataset is put into Table 1 and the RMSE Scores for the training & test are in 2.

Table 1: Classification Report for the Test Data

| Categories | precision | recall | f1-score | support | accuracy |
|---|---|---|---|---|---|
| ProbableAD | 0.82 | 0.8 | 0.81 | 35 | 0.82 |
| Control | 0.81 | 0.83 | 0.82 | 36 | |

Table 2: RMSE Scores

| Train RMSE | Test RMSE |
|---|---|
| 2.63 | 5.56 |

The Figure 1 displays the ROC curve for the Classification model and similarly, Appendix 3 shows the distribution of the residual for the test MMSE in a box plot.

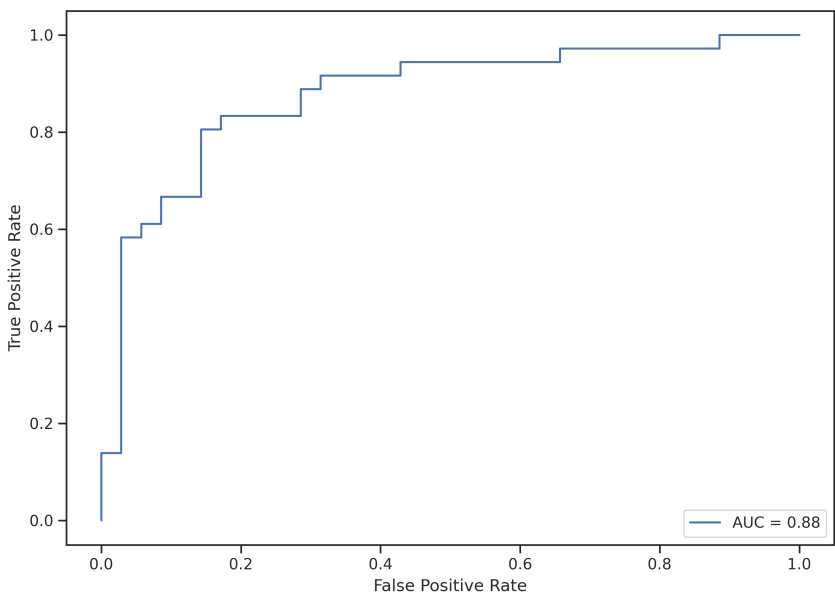

Figure 1: ROC Curve for the Classification Model

## 4 CONCLUSION

In summary, this paper provides an important contribution to the field of Alzheimer's Disease research by demonstrating the potential of MultiModal AutoML and feature extraction techniques to improve the detection and tracking of AD. The findings indicate that machines can effectively predict and evaluate AD with the right tools and techniques. While the study has some limitations due to the limited data, the results are still promising and warrant further investigation with larger and more diverse sample data. The development of accurate and cost-effective technologies to detect and track AD can significantly reduce the social and economic costs on families, making this research area of utmost importance for the future of healthcare.

URM STATEMENT

The authors acknowledge that all key authors of this work meet the URM criteria of ICLR 2023 Tiny Papers Track.

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

## A    APPENDIX

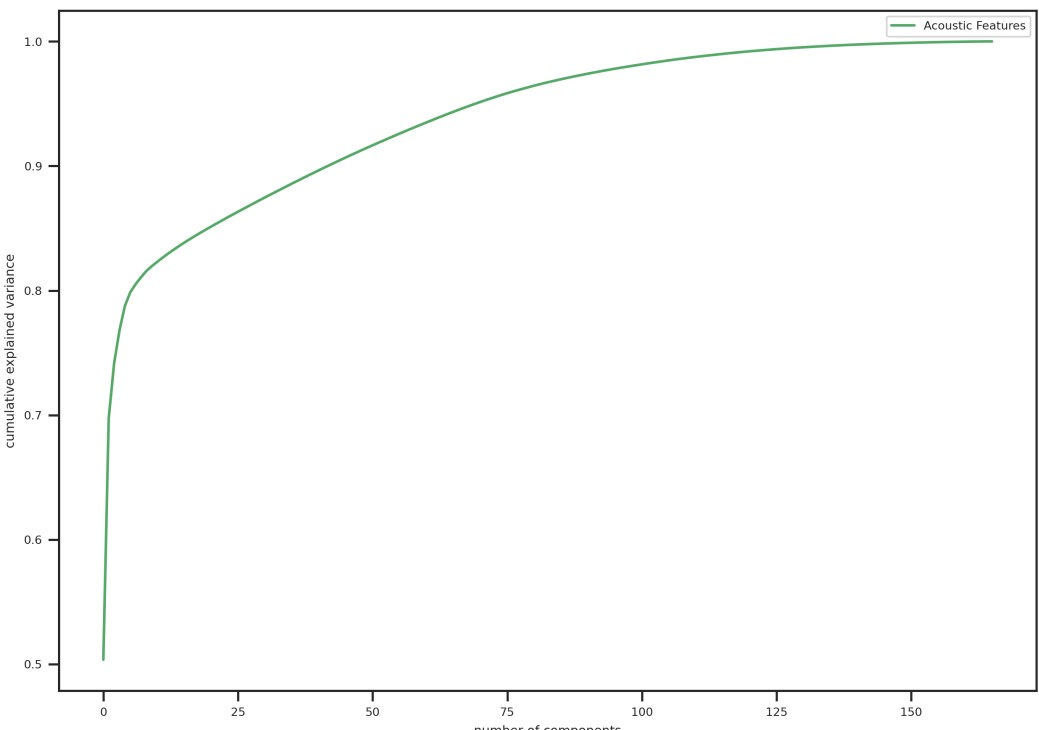

Figure 2: cumulative explained variance for acoustic features

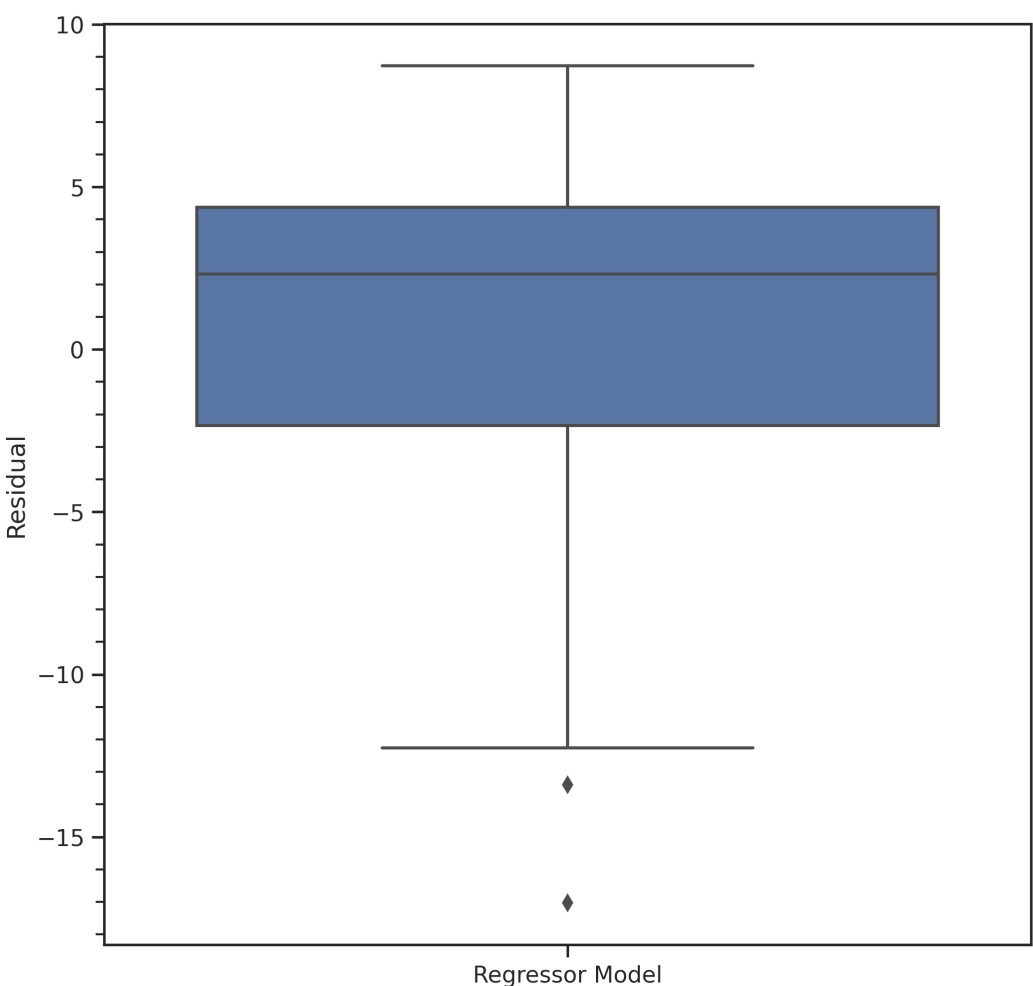

Figure 3: Residual Plot for the Regressor Model

