# OpenReview forum: "Experimenting with Multimodal AutoML: Detection and Evaluation of Alzheimer's Disease"
_ICLR.cc/2023/TinyPapers — Submitted to Tiny Papers @ ICLR 2023_

### Official Review · Reviewer_ghJz · 2023-03-29

**Confidence:** 5

**Summary Of Contributions:**

This paper demonstrates the potential of using multimodal AutoML to improve Alzheimer's disease detection. The authors achieved enhanced classification model performance and comparable regression model performance to the baseline, highlighting the effectiveness of combining various features and dimensionality reduction techniques in predicting and evaluating Alzheimer's Disease.

**Rating:**

Clear, Correct, and Reproducible (CCR): a submission which meets the reviewing criteria

**Strengths And Weaknesses:**

The authors demonstrate the potential of multimodal AutoMLframework, AutoGluon Tabular, in improving the detection and tracking of Alzheimer's Disease. The paper presents a clear methodology, results, and conclusion.

Strengths:

1. The paper addresses an important problem in healthcare, the Alzheimer's disease prediction and scoring of mental status using an AutoML framework.
2. The paper demonstrates an improvement in classification model performance and comparable performance in regression models to the baseline.
3. The paper is well-written and well-organized.

Weakness:

1. The authors should elaborate on the baseline model and methodology to provide a clearer context for comparison.
2. The paper could benefit from a more thorough discussion of the results, including a deeper analysis of the improvements in classification accuracy.
3. The authors could conduct an ablation study to show the benefit of the multimodal model, which would make it more convincing.

**Suggested Changes:**

1. The authors are encouraged to add more detail about the baseline and make the difference between it and the multimodal model more clear.
2. The authors are encouraged to add some discussion about the results.
3. The authors are encouraged to add some ablation study to show the benefit of the multimodal model.

---

### Meta-Review · Area_Chair_QwuJ · 2023-04-07

**Recommendation:** Invite to present
**Confidence:** 4

**Metareview:**

The paper adapts the widely talked about Multimodal models to Healthcare, particularly for detecting and tracking of Alzheimer's Disease.
The paper presents an experiment using AutoML, AutoGluon Tabular, to discover multimodal models for MMSE regression and AD detection. The findings on evaluation ADReSSo dataset indicate that machines can effectively predict and evaluate AD with the right tools and techniques as compared to other available baselines.

Overall, the paper is well-written and provides valuable insights into the use of AutoML and feature extraction techniques for AD detection and tracking. However, the paper could provide more details on the experimental setup, e.g. the baseline. Additionally, the paper could use some more discussion about the findings from the experiments and an ablation study to make the multimodal case. Minor typos standardizations could be fixed.

Based on the above, we would recommend this paper for acceptance with minor revisions.

**Summary:**

This paper presents an experiment using AutoML, AutoGluon Tabular, to discover multimodal models for MMSE regression and AD detection. The ADReSSo dataset was used to test the models, and the results showed improved performance in classification models and comparable performance in regression models to the baseline. The paper demonstrates the potential of MultiModal AutoML and feature extraction techniques to improve the detection and tracking of AD.

**Comments And Feedback To The Authors:**

We believe that paper is well-written and organized. It addresses an important problem in the ML for Healthcare domain with inspiration from existing approach that could be extended beyond detection and tracking of AD.

Here are some suggested improvements in the submission:
1. More but crisp details to the experimental design/setup, particularly the baseline model.
2. A brief discussion of results/conclusions from experiments could be added in the appendix to make the analysis wholistic.
3. A table corresponding to an ablation analysis for the available models could make the paper more convincing.
4. Proofread for minor typos and use macros wherever possible.

**Reason For Not Giving A Higher Recommendation:**

The paper is well-written and organized; However, it needs minor revisions (mentioned in the Comments/Feedback) to meet the Tiny Papers of CCR criteria.

**Reason For Not Giving A Lower Recommendation:**

N/A

---

### Decision · Program_Chairs · 2023-04-09

Invite to present